# 4-Propylphenol Alters Membrane Integrity in Fungi Isolated from Walnut Anthracnose and Brown Spot

**DOI:** 10.3390/jof11090610

**Published:** 2025-08-22

**Authors:** Xiaoli Yu, Shuhan Yang, Panhong Su, Haiyao Bi, Yaxuan Li, Xingxing Peng, Xiaohui Sun, Qunqing Wang

**Affiliations:** 1Yantai Academy of Agricultural Sciences, Yantai 265500, China; yuxiaoli92jnjn@126.com (X.Y.);; 2Yantai Key Laboratory of Wheat Resources Precise Evaluation and Biological Breeding, Yantai 265500, China; 3Department of Plant Pathology, College of Plant Protection, Shandong Agricultural University, Tai’an 271018, China; 4Qilu College, Shandong Agricultural University, Tai’an 271018, China

**Keywords:** *Colletotrichum*, *Alternaria*, 4-propylphenol, anthracnose, botanical fungicide, membrane disruption, phenology-driven, sustainable agriculture, disease management

## Abstract

Walnut anthracnose (*Colletotrichum gloeosporioides* and *C. siamense*) and brown spot (*Alternaria alternata*) cause severe yield losses globally. Conventional fungicides face the challenges of pathogen resistance and environmental toxicity. This study evaluates 4-propylphenol, a plant-derived phenolic compound, as an eco-friendly alternative against key fungal pathogens of walnut. In vitro assays determined EC_50_ values against target pathogens (29.11–31.89 mg·L^−1^) via mycelial growth inhibition and conidial germination suppression (EC_50_ = 55.04–71.85 mg·L^−1^). Mechanistic analyses confirmed membrane disruption through propidium iodide staining (9.5-to-14.0-fold fluorescence intensity increase), DNA leakage (77.82–85.15% at 250 mg·L^−1^), and protein efflux (58.10–66.49%). In field trials, we implemented a phenology-driven strategy: 100 mg·L^−1^ ground/canopy spray at flowering to reduce primary inoculum, followed by 400 mg·L^−1^ canopy application at fruiting. This protocol achieved 86.67% control efficacy against disease complexes with negligible phytotoxicity (SPAD variation < 5%). 4-propylphenol provides a sustainable solution through membrane-targeting action, effectively overcoming fungicide resistance in woody crops.

## 1. Introduction

Walnut (*Juglans regia* L.), a woody oilseed crop with significant economic value and ecological benefits, holds a prominent position in global agricultural trade due to its edible nuts and oil [1]. Fungal diseases have emerged as a critical constraint to sustainable walnut industrialization, with anthracnose representing one of the most devastating phytopathologies. The anthracnose pathogen complex (*Colletotrichum godetiae*, *C. gloeosporioides*, *C. fioriniae*, *C. nymphaeae*, and *C. siamense*) can infect up to 50% of fruits [2,3,4,5,6,7]. Disease onset typically occurs in early July, followed by exponential spread during August, culminating in over 35% infected fruits by early September [8]. Brown spot disease (alternatively termed brown apical necrosis), caused by *Alternaria* fungi, induces fruit browning, rot, and premature abscission, accounting for 30% of yield losses [9,10]. *Alternaria alternata* has been confirmed as the dominant pathogenic species in Shandong and Sichuan provinces of China [11,12].

Chemical control remains the predominant management strategy, but prolonged reliance on benzimidazoles and triazoles has precipitated a triple crisis: evolution of pathogen resistance [13], collateral damage to non-target organisms [14,15,16], and persistent environmental contamination [17]. This predicament underscores the urgent need for innovative green control technologies. Plant-derived fungicides, characterized by biodegradability, target specificity, and slower resistance development, have emerged as a pivotal strategy to overcome current control limitations [18,19,20]. Notably, existing research predominantly focuses on broad-spectrum antimicrobial secondary metabolites such as chitooligosaccharides [21], zedoary oil [22], and phenolic/terpenoid compounds [23], while systematic screening of plant-based agents against walnut-specific pathogens remains unexplored.

This study established a tripartite screening system for plant-derived fungicides against walnut fungal diseases, integrating activity-guided screening, membrane-targeted mechanism elucidation, and field-adaptive optimization. In vitro antimicrobial assays first revealed that 4-propylphenol exhibits significant inhibitory activity against Shandong’s predominant walnut pathogens: *Colletotrichum gloeosporioides* (EC_50_ = 31.89 ± 1.26 mg·L^−1^, half-maximal effective concentration), the newly identified *C. siamense* (EC_50_ = 31.06 ± 2.23 mg·L^−1^), and *A. alternata* (EC_50_ = 29.11 ± 1.88 mg·L^−1^). As a volatile plant secondary metabolite [24,25], this phenolic compound not only demonstrates previously reported antimicrobial spectra [26,27] but also disrupts pathogen membrane integrity [28,29], as mechanistically elucidated herein. A phenology-driven precision application protocol was implemented in field trials: whole-plant and ground spray (100 mg·L^−1^) during flowering to reduce primary inoculum, followed by canopy-directed application (400 mg·L^−1^) at the fruiting stage. This strategy achieved 86.67% control efficacy in combined anthracnose–brown spot incidence (*p* < 0.05), with chlorophyll SPAD analysis confirming negligible phytotoxicity (<5% variation vs. control). This work pioneers a comprehensive technical framework for plant-derived fungicide development in walnut protection, spanning in vitro screening, mechanistic validation, and field efficacy evaluation. The findings not only address critical gaps in eco-friendly walnut disease management but also establish a replicable paradigm for creating botanical pesticides in woody oilseed crops, advancing sustainable practices for specialty forest product industries.

## 2. Materials and Methods

### 2.1. Fungal Pathogens

The *C. gloeosporioides* strain (isolated from anthracnose-infected walnut fruits in Chongqing, China, 2019) was provided by Dr. Jian Gao (College of Plant Protection, Southwest University). The glyceraldehyde-3-phosphate dehydrogenase (*GAPDH*) gene was amplified with primers GDF1/GDR1 [30] and confirmed as *C. gloeosporioides* by BLASTn analysis (Query cover 100%, Ident 99.66% to GenBank accession KM053198.1).

The *A. alternata* strain (isolated from brown spot lesions on walnut fruits in Tai’an, Shandong, China, 2020) was provided by Prof. Hongyan Wang (College of Plant Protection, Shandong Agricultural University). The internal transcribed spacer (ITS) region was amplified with primers ITS4/ITS5 [31] and confirmed as *A. alternata* by BLASTn analysis (Query cover 100%, Ident 99.81% to GenBank accession OK036715.1).

The *C. siamense* strain HQ21 (GenBank accession: PV929873) was isolated from symptomatic anthracnose fruits in Tai’an, Shandong, China, 2021. Molecular characterization of *C. siamense* HQ21 is described in Section 2.2 and Section 2.3.

Partial sequences of molecular markers for all strains were deposited in Appendix A, including: the *GAPDH* gene fragment for *C. gloeosporioides*, the ITS region for *A. alternata*, and the *calmodulin* (*CAL*) gene fragment for *C. siamense* HQ21.

All strains were maintained on potato dextrose agar (PDA) slants at 4 °C in darkness with subculturing performed biannually.

### 2.2. Isolation and Identification of Pathogen HQ21 from Walnut Anthracnose Fruits

Symptomatic anthracnose fruits were collected and transported at 4 °C within 4 h. After surface sterilization with 75% ethanol (1 min), 1 cm^2^ tissue fragments from lesion margins were excised, treated with 1% NaClO (30 s), rinsed three times with sterile water, and cultured on PDA plates containing 100 mg·L^−1^ streptomycin (25 °C, 5–7 days in darkness). Pathogens were purified through three rounds of single-spore isolation on potato dextrose agar (PDA), yielding strain HQ21.

### 2.3. Pathogenicity Validation and Characterization of HQ21

**Pathogenicity assay:** Detached leaves of *Juglans regia* ‘Xiangling’ were needle-inoculated with HQ21 mycelial plugs (5 mm, 7-day-old PDA). Controls received sterile plugs. The leaves were maintained at 25 °C/85% RH for 5 days. Pathogens were re-isolated from lesions to fulfill Koch’s postulates.

**Morphological characterization:** Purified isolates were cultured on PDA plates at 25 °C for 7 days. Colony morphology and pigmentation were recorded, and conidioma dimensions were measured microscopically (Nexcope NE910-FLNingbo Yongxin Optics Co., Ltd., Ningbo, China, 400×).

**Molecular identification:** Genomic DNA was extracted using a modified CTAB protocol [32]. DNA purity was verified by the A260/A280 ratio (1.93) and A260/A230 ratio (1.86) using a NanoDrop 2000c spectrophotometer (Thermo Fisher Scientific, Wilmington, NC, USA). The *CAL* locus was amplified with primers CL1C (5′-GAATTCAAGGAGGCCTTCTC-3′) and CL2C (5′-CTTCTGCATCATGAGCTGGAC-3′), designed by Weir BS et al. [33], in 20 μL reactions containing 2× Rapid Taq Master Mix (Vazyme) 10 μL, Primers 0.5 μM each, and Template DNA 50 ng.

**Thermocycling parameters:** Amplifications were performed in a Bio-Rad T100 Thermal Cycler (Bio-Rad Laboratories, Inc., Hercules, CA, USA) with the following profile: initial denaturation at 94 °C (5 min); 35 cycles of 94 °C (30 s), 55 °C (45 s), and 72 °C (1 min); final extension at 72 °C (10 min). *CAL* DNA was purified using the SanPrep Column DNA Gel Extraction Kit (Sangon Biotech, Co., Ltd., Shanghai, China), and sequenced bidirectionally (Sangon Biotech, Shanghai, China).

**Phylogenetic reconstruction:** *CAL* sequences were BLASTN-searched against NCBI GenBank (GenBank accession: PV929873). Reference sequences from type strains were aligned using ClustalW in MEGA 11. The neighbor-joining tree was constructed with 1000 bootstrap replicates.

### 2.4. Dose–Response Analysis of 4-Propylphenol on Mycelial Growth

Based on preliminary screening of plant-derived phenolics against walnut pathogens, 4-propylphenol was selected for mechanistic and applied evaluation due to its superior antifungal potency and environmental safety profile. The compound (purity ≥ 98%, Sigma-Aldrich, St. Louis, MO, USA) was dissolved in methanol to prepare a 10,000 mg·L^−1^ stock solution (stored at 4 °C in darkness). Serial dilutions were performed to obtain amended PDA media with final concentrations of 20, 40, 60, 80, and 100 mg·L^−1^ (methanol concentration ≤ 1% *v*/*v*). Sterile methanol-amended PDA served as solvent control.

Mycelial plugs (5 mm diameter) from actively growing cultures of *C. gloeosporioides*, *C. siamense*, and *A. alternata* were inoculated onto amended plates. After 5 days of incubation at 25 °C in darkness, colony diameters were measured using the cross-hair method [34]. Mycelial growth inhibition was calculated asInhibition (%) = [(*D*_c_ − *D*_t_)/(*D*_c_ − 5)] × 100 where *D*_c_ and *D*_t_ represent colony diameters (mm) in control and treatment groups, respectively.

Probit regression analysis (SPSS 22.0) was employed to establish toxicological equations by correlating log-transformed concentrations (X) with inhibition probability units (Y). Half-maximal effective concentrations (EC_50_) and 95% confidence intervals were derived from the dose–response relationships.

### 2.5. Effects of 4-Propylphenol on Conidial Germination

**Conidial suspension preparation:** Conidia from *C. gloeosporioides*, *C. siamense*, and *A. alternata* were harvested by scraping sporulating cultures into 10 mL sterile water containing 0.05% Tween-80, vortexed for 30 s, and filtered through quadruple-layer sterile gauze. After centrifugation (4 °C, 3000× *g*, 10 min), pellets were resuspended and adjusted to 1 × 10^5^ CFU mL^−1^ using a hemocytometer.

**Germination inhibition assay:** 4-propylphenol was incorporated into 1.5% water agar (WA) at final concentrations of 15.625, 31.25, 62.5, 125, 250, and 500 mg·L^−1^. Aliquots (100 μL) of conidial suspensions were spread evenly on WA plates, with methanol-equivalent controls. Following 24 h incubation (25 °C, darkness), germination rates were assessed by counting germinated conidia. For each replicate, ≥50 spores were counted across five random fields (400×, Olympus CX23, Evident Corporation, Tokyo, Japan). Germination rate (%) = (Germinated spores/Total spores) × 100, where germination was defined by germ tube length ≥ spore diameter. Three replicates per treatment (total ≥ 150 spores) were evaluated.

### 2.6. Membrane Integrity Assessment

Membrane integrity was assessed by propidium iodide (PI) staining according to Stocks, S.M. [35]. Mycelial blocks of *C. gloeosporioides*, *C. siamense*, and *A. alternata* were cultured in potato dextrose broth (PDB) containing 40 mg·L^−1^ 4-propylphenol for 72 h at 25 °C. Hyphae were collected, washed thrice with PBS (pH 7.4), and stained with 10 µg·mL^−1^ propidium iodide (PI; Sigma-Aldrich) for 15 min. Control groups received equivalent 4-propylphenol-free PDB treatments. Membrane integrity was visualized under fluorescence microscopy (Nexcope NE910-FL; excitation 535 nm; emission 615 nm). Three hyphal segments per treatment were randomly selected for fluorescence intensity quantification using ImageJ v1.53.

### 2.7. Quantification of Mycelial DNA Leakage

Mycelial suspensions (5 mL, 20 mg·mL^−1^ dry weight) of *C. gloeosporioides*, *C. siamense*, and *A. alternata* were homogenously dispersed in 100 mL PDB containing 4-propylphenol (15.625, 31.25, 62.5, 125, and 250 mg·L^−1^) and incubated at 25 °C with shaking (120 rpm) for 24 h. After centrifugation (8000× *g*, 10 min, 4 °C), supernatants were filtered through 0.22 μm membranes. DNA leakage was quantified by measuring absorbance at 260 nm (A_260_) using a NanoDrop 2000c spectrophotometer (Thermo Fisher Scientific, Wilmington, NC, USA), with sterile PDB as blank control.DNA leakage (%) = (*A*_t_ − *A*_c_)/(*A*_m_ − *A*_c_) × 100

*A*_t_: Absorbance of treatment group;

*A*_c_: Absorbance of negative control (mycelia in 4-propylphenol-free PDB);

*A*_m_: Maximum leakage absorbance from mechanically disrupted mycelia (sonicated at 20 kHz for 5 min).

Three replicates per group were analyzed via nonlinear regression (SPSS 22.0).

### 2.8. Protein Leakage from Conidia

Hemocytometer-calibrated conidial suspensions (1 × 10^5^ conidia·mL^−1^) of *C. gloeosporioides*, *C. siamense*, and *A. alternata* were aliquoted (5 mL) into 100 mL PDB containing 4-propylphenol (15.625–250 mg·L^−1^, 2-fold dilution series). Following shaking incubation (25 °C, 150 rpm, 24 h), samples were centrifuged (8000× *g*, 10 min, 4 °C). Filtered supernatants (0.22 μm) were analyzed at 280 nm (A_280_) using a NanoDrop 2000c.Protein leakage (%) = (*A*_t_ − *A*_c_)/(*A*_m_ − *A*_c_) ×100

*A*_t_: Absorbance of treated conidia;

*A*_c_: Absorbance of untreated control (conidia in 4-propylphenol-free PDB);

*A*_m_: Maximum leakage from sonicated conidia (20 kHz, 5 min).

Triplicate biological replicates were subjected to nonlinear regression analysis (SPSS 22.0).

### 2.9. Leaf Treatment and Pathogen Inoculation

Mature detached leaves (30 ± 2 days old) of *Juglans regia* ‘Xiangling’ were collected from field-grown trees. Three biological replicates per treatment group were established, with each replicate consisting of one leaf. Leaves were treated with 4-propylphenol solutions (15.625–500 mg·L^−1^, containing 0.1% Tween-80 surfactant) via foliar spray. Negative controls received sterile water with 0.1% Tween-80. After air-drying, each leaf was needle-inoculated at midribs with 5 mm mycelial plugs of *C. siamense* (7-day-old cultures) and incubated in a humidity chamber (25 °C, 90% RH) for 7 days. To enhance lesion visibility by removing chlorophyll, leaves were decolorized by immersion in 75% (*v*/*v*) aqueous ethanol within a water bath maintained at 60 °C until complete chlorophyll removal was achieved. After rinsing with distilled water and blotting dry, lesion diameters (mm) were measured.Inhibition (%) = (*D*_c_ − *D*_t_)/*D*_c_ × 100, where *D*_c_ and *D*_t_ denote lesion diameters (mm) in control and treated groups, respectively. Data from three biological replicates were pooled, and dose–response curves were fitted using SPSS 22.0 to determine EC_50_ values and 95% confidence intervals. Significant differences were analyzed via one-way ANOVA and Duncan’s test (α = 0.01).

### 2.10. Field Safety Evaluation

Field trials were conducted on 5 June 2022 (early fruit expansion stage) using ten-year-old *Juglans regia* ‘Xiangling’ trees with uniform growth characteristics. A randomized complete block design was employed with three replicates. Each replicate consisted of one experimental tree, serving as an individual block, resulting in three total trees. On each tree, twelve branches with uniform growth status (similar age, orientation, and diameter) were selected and randomly allocated into four treatment groups, with three branches per group. Treatments consisted of three concentrations of 4-propylphenol (100, 200, and 400 mg·L^−1^) and a water-treated control. Directional foliar spraying was applied specifically to the assigned branch groups using a knapsack electric sprayer (0.3 MPa pressure, 0.4 mm nozzle diameter). The spray volume was calibrated to 500 L·ha^−1^ based on pre-trial optimization, ensuring complete leaf coverage on treated branches without runoff and implementing physical isolation to prevent spray drift between treatments. At 30 days post-treatment, chlorophyll SPAD values were measured on a total of 30 mature leaves per treatment group (i.e., 10 leaves/replicate) using a handheld SPAD-502 m (Konica Minolta, Inc., Tokyo, Japan). Significant differences among treatments were determined by one-way ANOVA followed by Duncan’s multiple range test (α = 0.05) in SPSS 22.0.

### 2.11. Optimization of Field Application Strategies

Given that the key pathogens causing walnut fruit diseases—*C. gloeosporioides* and *C. siamense* (anthracnose), and *A. alternata* (brown spot)—primarily infect fruits and collectively result in direct economic losses, this trial shifted the evaluation focus from leaf lesions to comprehensive fruit disease assessment. A randomized block trial (3 replicates) was conducted in 2023 at Huangqian Town’s standardized walnut orchard using *Juglans regia* ‘Xiangling’ trees. Three treatment regimens were evaluated (Table 1): T1: Initial infection control (100 mg·L^−1^ canopy spray at flowering; water spray at fruiting); T2: T1 protocol + ground spray (100 mg·L^−1^ canopy + ground application at flowering; water spray at fruiting); T3: T2 protocol + enhanced fruiting-stage treatment (400 mg·L^−1^ canopy spray at fruiting); CK: Water-only controls for both stages.

Applications were performed using a knapsack electric sprayer (0.3 MPa, 0.4 mm nozzle) with ~500 L·ha^−1^ volume for full coverage. n = 3 biological replicates per treatment.

Disease assessment: Fifteen days post-treatment, 30 fruits per group were graded:

0: No lesions;

1: 1–2 lesions;

2: 3–5 lesions;

3: >5 lesions or coalesced necrosis.Disease index (*DI*) = Σ(*Disease grade* × *fruit count*)/(3 × *total fruits*) × 100.Control efficacy (%) = (*DI*_CK_ − *DI*_treatment_)/*DI*_CK_ × 100.

Data were analyzed using SPSS 22.0 with significance thresholds at α = 0.05.

## 3. Results

### 3.1. Isolation, Identification, and Pathogenicity of Colletotrichum siamense in Huangqian Walnut Orchards

Walnut fruits exhibiting anthracnose symptoms were collected from Huangqian Town, Tai’an City (Figure 1A). Strain HQ21, isolated through single-spore purification, formed circular colonies (78.3 ± 1.5 mm diameter) on PDA after 5 days at 25 °C. Colonies displayed dense aerial mycelia with dark gray centers and cream-colored margins on the obverse side (Figure 1B), while reverse sides exhibited gray-green central pigmentation (Figure 1C). Hyphae appeared cylindrical with septa (Figure 1D), producing hyaline, aseptate conidia measuring (14.2 ± 1.3) × (4.2 ± 0.6) μm (Figure 1E). Pathogenicity assays induced characteristic anthracnose lesions on inoculated leaves within 5 days, mirroring field symptoms (Figure 1F), while the controls remained asymptomatic (Figure 1G). Re-isolation from diseased tissues yielded morphologically identical colonies to HQ21, fulfilling Koch’s postulates. Phylogenetic analysis of concatenated *CAL* sequences placed HQ21 within a highly supported *C. siamense* clade (Bootstrap = 98%; genetic distance ≤ 0.002) (Figure 2). Integrated morphological and molecular evidence confirmed HQ21 as *C. siamense* Prihastuti, L. Cai & K.D. Hyde. The *CAL* gene sequence is available in GenBank under accession number PV929873.

### 3.2. Virulence Analysis of 4-Propylphenol Against Target Pathogens

4-propylphenol exhibited significant dose-dependent inhibition of mycelial growth across all three pathogens. Serial dilutions (20–100 mg·L^−1^) demonstrated significant dose–response relationships across all pathogens (R^2^ = 0.827–0.948), with EC_50_ values of 31.893, 31.063, and 29.111 mg·L^−1^ for *C. gloeosporioides*, *C. siamense*, and *A. alternata*, respectively. Steep concentration–response profiles were evidenced by toxicity equation slopes (3.02–3.81) (Table 2 and Figure 3).

### 3.3. Inhibitory Effects of 4-Propylphenol on Conidial Germination

As shown in Table 3 and Figure 4, 4-propylphenol exhibited significant concentration-dependent inhibition (*p* < 0.01) on conidial germination across all three pathogens. After 24 h treatment, the germination rates for *C. gloeosporioides*, *C. siamense*, and *A. alternata* were 80.00 ± 2.23%, 84.75 ± 2.91%, and 85.39 ± 3.18%, respectively. At 500 mg·L^−1^, the germination rates decreased to 5.4 ± 1.7%, 7.5 ± 1.47%, and 9.2 ± 0.5%, corresponding to 89.22–93.25% suppression compared to controls. The EC_50_ values for conidial germination inhibition were 55.037, 71.854, and 64.414 mg·L^−1^ for *C. gloeosporioides*, *C. siamense*, and *A. alternata*, respectively.

### 3.4. Membrane Disruption Effects of 4-Propylphenol on Target Pathogens

Propidium iodide (PI) staining revealed significant membrane damage in 4-propylphenol-treated hyphae (Figure 5). The treatment groups exhibited intense red fluorescence signals (fluorescence intensity: *C. gloeosporioides* 55.74 ± 3.04 a.u., *C. siamense* 55.18 ± 6.54 a.u., and *A. alternata* 61.01 ± 8.13 a.u.), which were markedly higher than those in untreated controls (*C. gloeosporioides*: 3.98 ± 1.41 a.u.; *C. siamense*: 4.01 ± 1.00 a.u.; *A. alternata*: 6.45 ± 5.68 a.u.) (*p* < 0.001), representing 14.0-fold fluorescence intensity increases for *C. gloeosporioides*, 13.8-fold for *C. siamense*, and 9.5-fold for *A. alternata* compared to untreated controls. These results confirm that 4-propylphenol induces significant alterations in membrane permeability.

### 3.5. Dose-Dependent Induction of DNA Leakage by 4-Propylphenol

As illustrated in Figure 6, 4-propylphenol significantly increased DNA leakage from *C. gloeosporioides*, *C. siamense*, and *A. alternata* hyphae in a concentration-dependent manner (R^2^ = 0.802–0.924). At 250 mg·L^−1^, DNA leakage reached 77.82–85.15% of maximum saturation values achieved through mechanical disruption.

### 3.6. Protein Leakage Induced by 4-Propylphenol

As demonstrated in Figure 7, 4-propylphenol significantly compromised fungal membrane permeability, inducing dose-dependent protein leakage. Leakage levels exhibited strong positive correlations with compound concentrations (*R*^2^ = 0.974–0.991). At 250 mg·L^−1^, protein leakage reached 58.10–66.49% of maximum saturation values: *C. gloeosporioides* (65.02 ± 0.53%), *C. siamense* (58.10 ± 0.44%), and *A. alternata* (66.49 ± 1.05%).

### 3.7. Protective Effects of 4-Propylphenol Against C. siamense Infection in Detached Walnut Leaves

Detached walnut leaves treated with 4-propylphenol (15.625–500 mg·L^−1^) showed significant concentration-dependent suppression of *C. siamense* infection (R^2^ = 0.99903). Visual assessment of disease progression revealed distinct phenotypic differences across treatment concentrations (Figure 8A), with complete inhibition of pathogen colonization observed at the highest concentration of 500 mg·L^−1^. The dose–response relationship demonstrated progressive efficacy, achieving 47.52 ± 3.97% disease inhibition at 62.5 mg·L^−1^, 84.67 ± 3.25% at 250 mg·L^−1^, and 100% at 500 mg·L^−1^ (Figure 8C). Quantitative analysis through probit modeling determined the half-maximal effective concentration (EC_50_) as 96.54 mg·L^−1^ (Figure 8B). Statistical comparison of inhibition rates across concentrations revealed significant differential efficacy (*p* < 0.01, Duncan’s test), with distinct effectiveness groupings indicated by letter annotations in the histogram (Figure 8C). This comprehensive analysis confirms 4-propylphenol’s potent capacity to protect foliar tissues against anthracnose pathogenesis, achieving complete pathogen suppression at the maximum tested concentration.

### 3.8. Field Safety Evaluation of 4-Propylphenol on Walnut Trees

Field trials evaluating the physiological safety of different 4-propylphenol concentrations on walnut trees demonstrated the following results (Figure 9): Foliar applications at 100, 200, and 400 mg·L^−1^ maintained leaf SPAD values at 43.6 ± 1.18, 44.4 ± 1.87, and 43.84 ± 1.96, respectively (Figure 9A). Compared to the water-treated control group (44.55 ± 1.39; Figure 9B), no significant differences (*p* > 0.05) in SPAD values were observed across treatments. Moreover, the 400 mg·L^−1^ treatment group (Figure 9C) exhibited no visible phytotoxicity symptoms. Based on comprehensive safety evaluation parameters (SPAD stability and foliar phenotypic integrity) and the potential enhancement of disease control efficacy at higher concentrations, 400 mg·L^−1^ is recommended as the optimal application concentration to balance phytological safety and field efficacy.

### 3.9. Integrated Application Strategies for Disease Control Efficacy

As shown in Figure 10E, 4-propylphenol demonstrated significant field efficacy against walnut fruit diseases. The T1 treatment (100 mg·L^−1^ canopy spray at flowering + water spray at fruiting; Figure 10B) achieved 47.78 ± 5.09% control efficacy with 28.89% diseased fruits. The T2 treatment (100 mg·L^−1^ canopy + ground spray at flowering + water spray at fruiting; Figure 10C) significantly enhanced overall efficacy to 80.00 ± 6.67% (*p* < 0.01; Figure 10E), reducing disease incidence to 12.22%. The T3 treatment (T2 + 400 mg·L^−1^ canopy spray at fruiting; Figure 10D) achieved optimal performance with 7.78% diseased fruits and 86.67 ± 3.33% control efficacy, significantly surpassing T1/T2 (*p* < 0.01; Figure 10E). Untreated controls (CK, Figure 10A) exhibited 56.67% disease incidence (Figure 10E).

## 4. Discussion

The pathogen diversity of walnut anthracnose and brown spot, coupled with their cross-host transmission characteristics, poses a persistent threat to the global walnut industry. Over 10 species of *Colletotrichum* spp. have been confirmed to infect walnuts [36,37,38,39], among which *C. siamense* has emerged as a critical pathogen due to its broad host adaptability and geographical dispersal capacity. The pathogenic strain of *C. siamense* identified in Tai’an, Shandong Province, further validates the expansion of this species in temperate woody crops. Notably, *C. siamense* exhibits a host range spanning at least 15 genera across seven plant families, including *Cornus* spp. [40], *Ixora chinensis* [41], *Persea Americana* [42], and *Citrus* spp. [43]. This polyphagous trait suggests potential cross-species transmission through host plant networks within agroecosystems, significantly amplifying the epidemiological complexity and control challenges of the disease.

This study is the first to elucidate the molecular mechanism by which 4-propylphenol exerts antifungal effects via targeted disruption of pathogen membrane integrity. Fluorescence staining revealed significant membrane damage, evidenced by 9.5-to-14.0-fold increases in propidium iodide (PI) staining intensity following treatment, consistent with the classical mode of phenolic compounds that alter membrane potential to induce ion homeostasis imbalance [44,45]. Unlike conventional fungicides (e.g., triazoles targeting enzyme inhibition) [46,47], this nonspecific membrane-targeting mechanism may theoretically reduce the risk of resistance development [48]. However, potential fungal adaptive mechanisms, such as MAPK signaling pathway-mediated dynamic regulation of membrane composition, warrant further investigation [49,50].

In bioactivity assessments, 4-propylphenol demonstrated EC_50_ values of 29.11–31.89 mg·L^−1^ against plant pathogens, comparable to the optimal eugenol derivative against *Botrytis cinerea* (EC_50_ = 31 mg·L^−1^) [51] and superior to 1-sulfonyloxy/acyloxy eugenol analogs against *Phytophthora capsica* (EC_50_ = 70.8 mg·L^−1^) [52]. Notably, its inhibition of conidial germination (EC_50_ = 55.04–71.85 mg·L^−1^) exhibited approximately 55-fold greater efficacy than ginseng stem-leaf total saponins (EC_50_ = 3037.7 mg·L^−1^) [53] and outperformed commercial biofungicides such as *Bacillus subtilis* lipopeptides (EC_50_ > 100 mg·L^−1^) [54]. The compound also displayed rapid (>90% inhibition within 24 h) and sustained antifungal activity, highlighting its unique advantages in plant disease management systems.

The proposed gradient application strategy (100 mg·L^−1^ ground spray (canopy and soil application) at the flowering stage; 400 mg·L^−1^ targeted canopy spraying at the fruiting stage) demonstrated high field efficacy in disease control while aiming at suppressing primary inoculum sources. This approach shares conceptual similarities with grape anthracnose management protocols (e.g., copper-based agents at bud break and microbial agents at flowering) [55], yet simplifies operational complexity through optimized concentration gradients of a single active ingredient. However, soil drenching may exert non-target effects on microbial communities. Metagenomic analysis is recommended to systematically evaluate impacts on soil microbial network structure and functional gene expression, thereby informing the development of ecologically sustainable application protocols.

## 5. Conclusions

This study established an integrated system encompassing membrane-targeting mechanism analysis and phenology-adapted application technology, providing an innovative framework for sustainable disease management in woody oil crops. Key findings include the following: (1) Novel antimicrobial mechanism: 4-propylphenol triggers cellular content leakage through membrane lipid disruption, demonstrating a resistance-avoidance mode that circumvents the mutation risks inherent in conventional fungicides targeting specific molecular sites. (2) Application strategy innovation: A hierarchical management strategy, integrating flowering-stage ground disinfection and fruit-development-stage canopy protection, showed significant efficacy against walnut anthracnose, enabling chemical input reduction while maintaining eco-friendly orchard operations. Future research should prioritize three directions: molecular-level characterization of 4-propylphenol’s specific binding sites with membrane lipids, establishment of regional pathogen resistance surveillance networks, and development of nano-encapsulated formulations to enhance residual activity. This integrated approach not only addresses critical challenges in walnut disease control but also provides replicable solutions for sustainable cultivation of other woody oil crops, including Chinese hickory (*Carya cathayensis*) and oil-tea camellia (*Camellia oleifera*). Through mechanism-driven technological innovation, this work achieves effective synergy between ecological safety and pathogen control, advancing the development of green control technologies in agricultural phytopathology.

## Figures and Tables

**Figure 1 jof-11-00610-f001:**
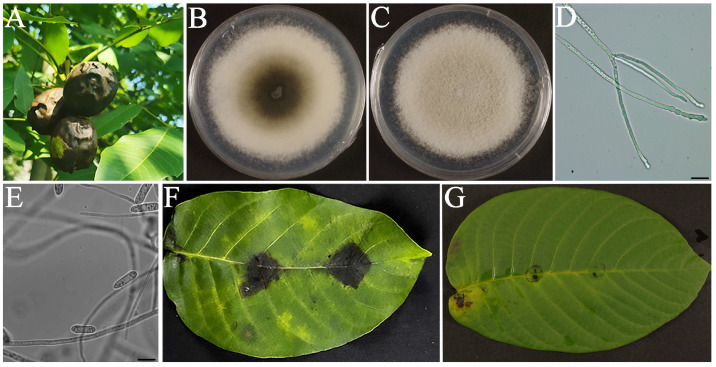
Isolation and pathogenicity determination of walnut anthracnose pathogen. (**A**): Field symptoms of walnut anthracnose on fruits. (**B**,**C**): Morphology of strain HQ21 on PDA, 5 days after inoculation (front and reverse view). (**D**): Hypha structure. (**E**): Conidia. (**F**): Disease progression in artificial inoculation assay (5 dpi). (**G**): Control leaf on PDA block inoculation. Scale bars: (**D**,**F**) = 20 μm.

**Figure 2 jof-11-00610-f002:**
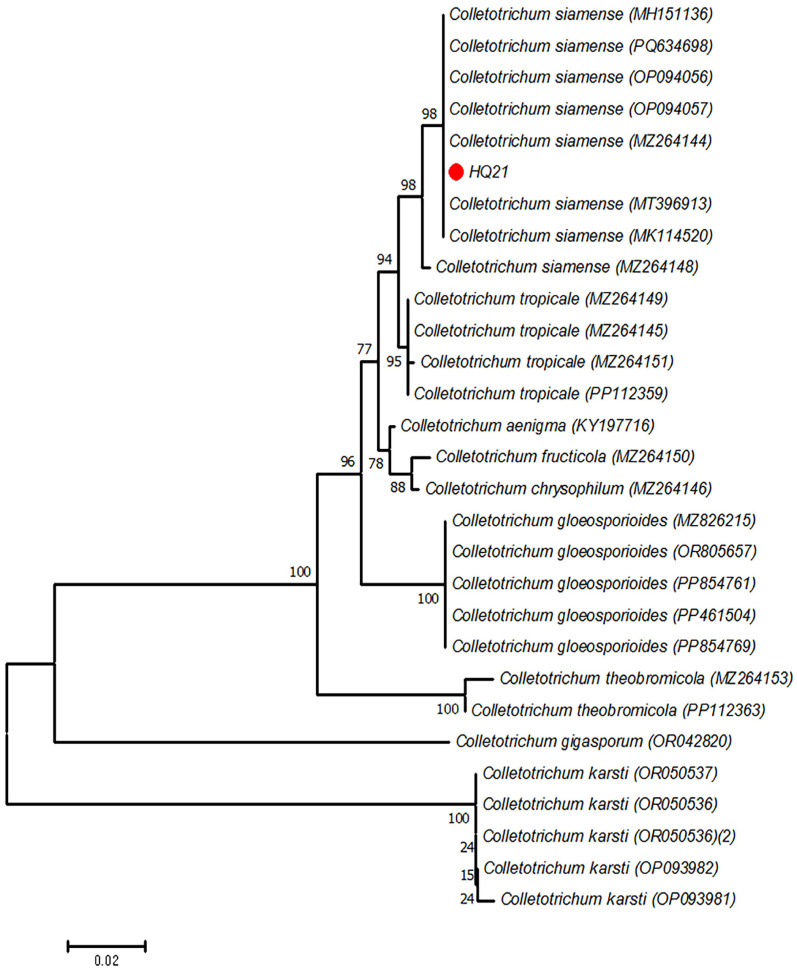
Phylogenetic analysis of strain HQ21 based on *CAL* gene sequences. The neighbor-joining tree was constructed using MEGA 11 software with 1000 bootstrap replicates. Sequences of reference strains (indicated by GenBank accession numbers) were retrieved from NCBI.

**Figure 3 jof-11-00610-f003:**
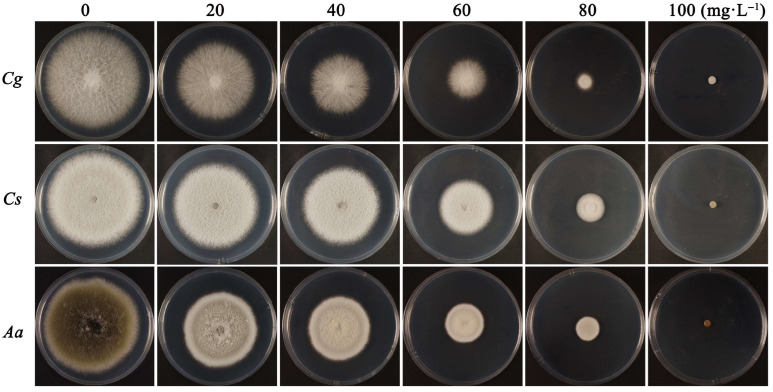
Dose-dependent inhibition of 4-propylphenol (0–100 mg·L^−1^) on mycelial growth of *C. gloeosporioides* (Cg), *C. siamense* (Cs), and *A. alternata* (Aa).

**Figure 4 jof-11-00610-f004:**
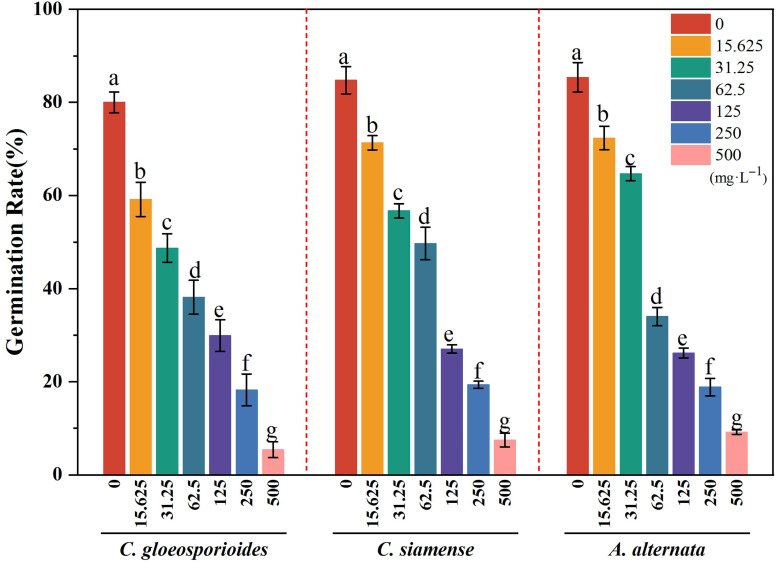
Inhibitory effects of 4-propylphenol on conidial germination of three pathogens. Column colors represent concentrations: 0, 15.625, 31.25, 62.5, 125, 250, and 500 mg·L^−1^. Different lowercase letters indicate significant differences (*p* < 0.01, Duncan’s multiple range test).

**Figure 5 jof-11-00610-f005:**
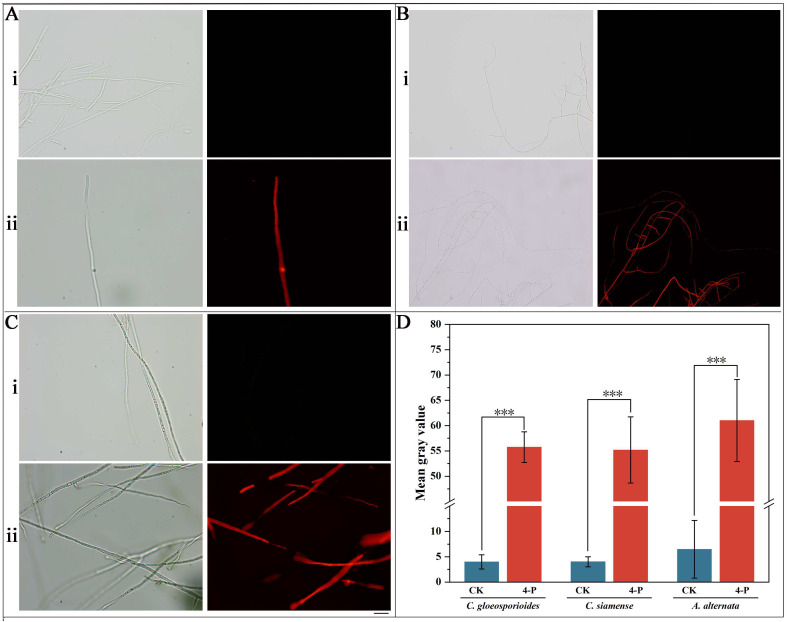
Effects of 4-propylphenol on membrane integrity of three pathogens. (**A**): *C. gloeosporioides* hyphae under bright-field (*left*) and PI fluorescence (**right**): (i) untreated control; (ii) 40 mg·L^−1^ 4-propylphenol treatment. (**B**): *C. siamense* hyphae under bright-field (*left*) and PI fluorescence (**right**): (i) untreated control; (ii) 40 mg·L^−1^ 4-propylphenol treatment. (**C**): *A. alternata* hyphae under bright-field (*left*) and PI fluorescence (**right**): (i) untreated control; (ii) 40 mg·L^−1^ 4-propylphenol treatment. (**D**): Quantitative analysis of PI fluorescence intensity showing 9.5-to-14.0-fold increases in treated hyphae. Scale bar = 200 μm (applicable to (**A**–**C**)). *** denote significant differences (*p* < 0.001, Duncan’s multiple range test).

**Figure 6 jof-11-00610-f006:**
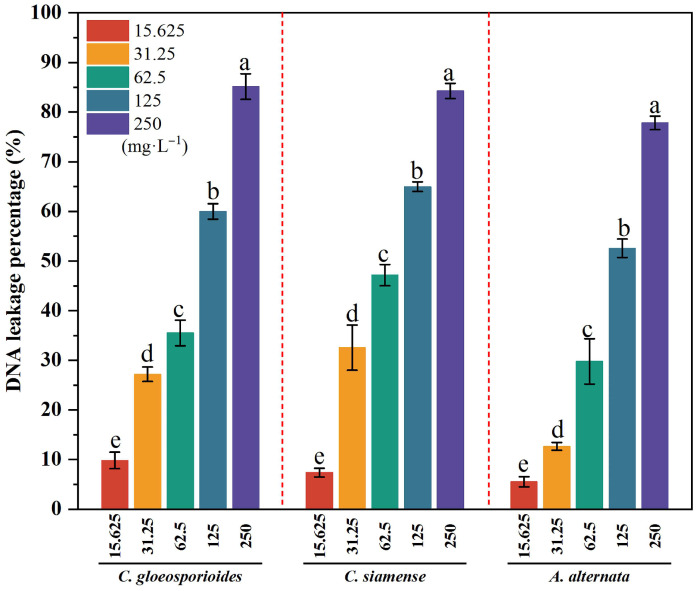
Dose-effect of 4-propylphenol on DNA leakage of three pathogens. Column colors represent concentrations: 15.625, 31.25, 62.5, 125, and 250 mg·L^−1^. Different lowercase letters denote significant differences (*p* < 0.01, Duncan’s multiple range test).

**Figure 7 jof-11-00610-f007:**
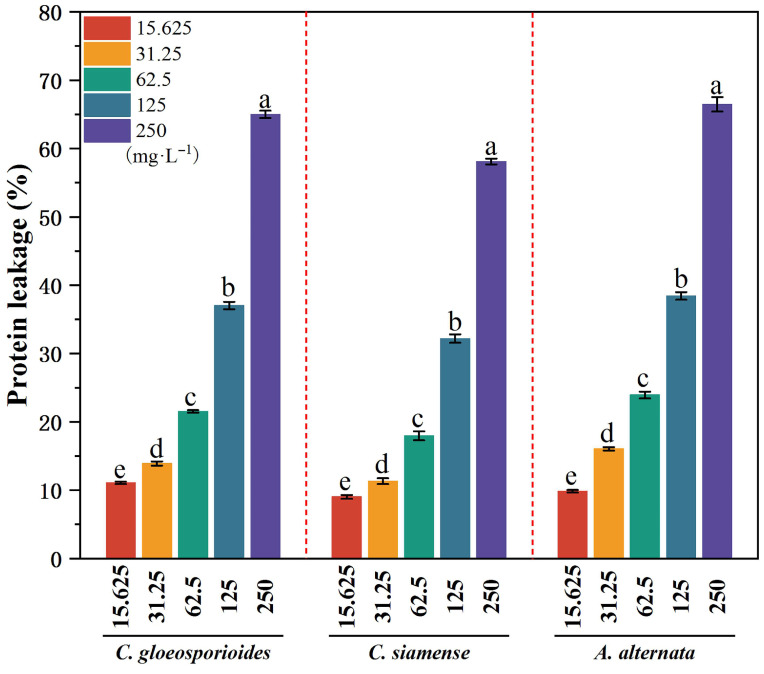
Dose-dependent protein leakage induced by 4-propylphenol in three pathogens. Column colors represent concentrations: 15.625, 31.25, 62.5, 125, and 250 mg·L^−1^. Different lowercase letters indicate significant differences (*p* < 0.01, Duncan’s multiple range test).

**Figure 8 jof-11-00610-f008:**
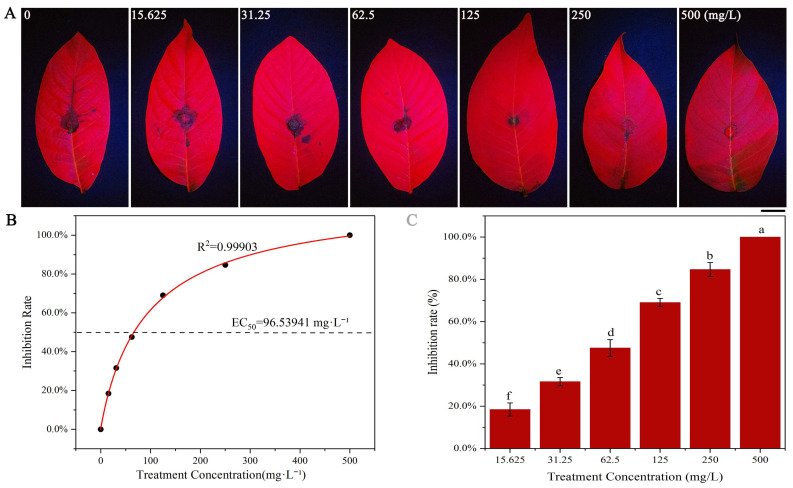
Protective effect of 4-propylphenol against *C. siamense* infection on detached walnut leaves. (**A**): Phenotypic comparison of detached walnut leaves treated with 4-propylphenol at different concentrations (0, 15.625, 31.25, 62.5, 125, 250, and 500 mg·L^−1^) and inoculated with *C. siamense*. Images were captured 5 days post-inoculation after chlorophyll removal via ethanol decolorization. Scale bar = 2 cm. (**B**): Dose–response curve of *C. siamense* inhibition by 4-propylphenol. (**C**): Histogram of inhibition rates at tested concentrations. Columns labeled with different lowercase letters show statistically significant differences (*p* < 0.01, Duncan’s multiple range test).

**Figure 9 jof-11-00610-f009:**
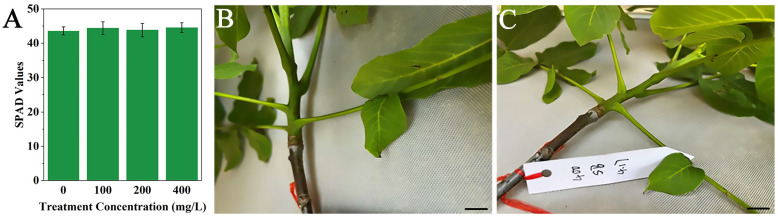
Evaluation of 4-propylphenol phytotoxicity in field-grown walnut trees (*Juglans regia*). (**A**): Leaf chlorophyll content (SPAD values) across experimental treatments (mean ± SD, n ≥ 30 leaves/treatment). (**B**): Representative foliage from water-treated control plants. (**C**): Foliar phenotype following 400 mg·L^−1^ 4-propylphenol application. Scale bar = 5 cm (applicable to (**B**,**C**)).

**Figure 10 jof-11-00610-f010:**
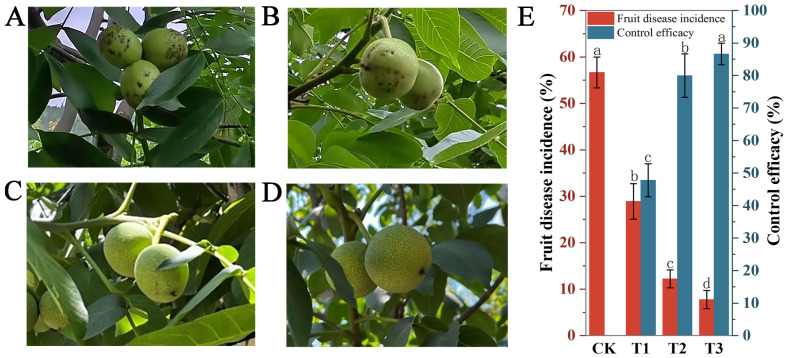
Field efficacy of 4-propylphenol application strategies against walnut diseases. (**A**). Untreated control (CK; water spray at both stages). (**B**). T1 treatment (100 mg·L^−1^ flowering canopy spray). (**C**). T2 treatment (100 mg·L^−1^ flowering canopy + ground spray). (**D**). T3 treatment (T2 + 400 mg·L^−1^ fruiting canopy spray). (**E**). Comparative disease incidence and control efficacy across treatments. Different lowercase letters denote significant differences (*p* < 0.01, Duncan’s multiple range test).

**Table 1 jof-11-00610-t001:** Field application strategies of 4-propylphenol.

Treatment	May 5 (Flowering)	June 10 (Fruiting)
T1	100 mg·L^−1^ canopy	Water spray
T2	100 mg·L^−1^ canopy + ground	Water spray
T3	100 mg·L^−1^ canopy + ground	400 mg·L^−1^ canopy
CK	Water spray	Water spray

**Table 2 jof-11-00610-t002:** Virulence regression parameters of 4-propylphenol against walnut pathogens.

Pathogen	Toxicity Regression Equation	*R* ^2^	EC_50_ (mg·L^−1^)	95% Confidence Interval (mg·L^−1^)
*C. gloeosporioides*	Y = −5.69 + 3.81X	0.827	31.89	8.16–49.25
*C. siamense*	Y = −4.80 + 3.22X	0.948	31.06	17.75–41.67
*A. alternata*	Y = −4.41 + 3.02X	0.882	29.11	9.52–42.63

Y: Inhibition probability units; X: Log-transformed concentrations. EC_50_ values derived from Probit analysis.

**Table 3 jof-11-00610-t003:** Virulence regression parameters of 4-propylphenol against walnut pathogen conidial germination.

Pathogen	Toxicity Regression Equation	*R* ^2^	EC_50_ (mg·L^−1^)	95% Confidence Interval (mg·L^−1^)
*C. gloeosporioides*	Y = −2.31 + 1.33X	0.966	55.037	44.148–67.138
*C. siamense*	Y = −2.82 + 1.52X	0.983	71.854	60.220–85.194
*A. alternata*	Y = −2.72 + 1.49X	0.956	64.414	41.959–93.571

Y: Inhibition probability units; X: Log-transformed concentrations. EC_50_ derived from Probit analysis.

## Data Availability

The data presented in this study are openly available in Zenodo, reference number 16245946.

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
