# Peer review of "4-Propylphenol Alters Membrane Integrity in Fungi Isolated from Walnut Anthracnose and Brown Spot"

_jof, 2025, doi:10.3390/jof11090610_

Round 1

Reviewer 1 Report

The manuscript presents interesting results for the scientific community, particularly in the field of phytopathology. The use of compounds alternative to azoles is of particular interest for reducing the presence of resistant phytopathogens. However, there are some areas of opportunity that need improvement.

For example, only include the compound that was evaluated and is mentioned in the title; otherwise, you will have to perform all the assays for all compounds. I understand that you chose the compound that produced the greatest mycelial inhibition; however, inhibition can vary in spore germination. The same can happen with the effects on the mycelium. Can you assure me that linalool, which inhibits 46%, doesn’t affect mycelial morphology more than 4-propylphenol? Could you please explain why the decisions in the virulence assays with 4-ethylphenol were based solely on mycelial growth? In any case, you should have selected based on spore germination. All these unknowns can be resolved by focusing solely on 4-propylphenol or by including all assays for the other compounds, at least for those that achieved over 50% inhibition.

Please respond to and address each point observed, and indicate the modification made in the revised document.

Title: The following title is recommended: 4-Propylphenol alters the membranes of fungi isolated from anthracnose and walnut brown spot.

Materials and methods

Line 81: expand the description of how the isolates were obtained and how they were transported to the laboratory. What taxonomic descriptors were used?

Line 84: provide more information about the isolate provided by Dr. Gao—where it was isolated, how it was identified, etc.

Line 85: Indicate what the identical preservation conditions are.

Line 85: provide additional information about the isolate provided by Professor Hongyan—where it was isolated, how it was identified, etc.

Line 106: Were the cultures shaken? How many revolutions per minute were the cultures shaken?

Line 108: Please provide the reference for the modified CTAB protocol or describe it in detail.

Line 113: Provide more data about the thermocycler.

Line 121: In the title, only 4-propylphenol is mentioned, so the other compounds should be removed from the document.

Line 130: cite the source of the method, as it is reported in another article.

Line 155: How many spores did they count? How many spores correspond to 100%?

Section 2.7: You must indicate the source or reference for the technique. All protocols must indicate the source or bibliographic reference, unless it is a new, unreported technique; however, that is not the case.

Figure 1 shows that the scale appears to be incorrect; please verify and correct it. The scale bar indicates 10 microns; however, in my experience, the length of a Alternaria spore exceeds 10 microns.

How can you verify the quality and integrity of the DNA? Do you have photographic evidence of the gels? What absorbance ratios were obtained on the NanoDrop for the identified isolate?

The obtained sequence could not be verified; please check the GenBank accession number. Or include the sequence in supplementary material.

Reviewer 2 Report

This is a valuable manuscript, contributing new advances to science. The authors conducted multi-stage experiments. Overall, the experimental methodology is presented in a satisfactory manner. However, in some places, additional information is needed, as certain details and important data are omitted. For example, it is not specified whether detached leaves or attached leaves (e.g., seedlings in pots) were inoculated (see Remarks). It is difficult to expect appropriate host plant defense responses on detached leaves. Some important data, instead of being presented in the M&M section, are only disclosed in the Results section. The results are presented very clearly; subsequent experiments are presented systematically. C. siamense was identified on a morphological and phylogenetic basis. In addition, the following are presented: (i) antimicrobial activity of plant-derived metabolites in vitro, (ii) virulence and inhibitory analysis of 4-Propylphenol against C. gloeosporioides, C. siamense, and A. alternata, (iii) induction of DNA leakage by 4-Propylphenol, and (iv) protective effects of 4-Propylphenol against C. siamense. This study is the first to demonstrate the mechanism by which 4-propylphenol induces antifungal effects (disruption of pathogen membrane integrity). The conclusion is correct, based on the conducted research. After considering the comments included in the Remarks (minor revision), the manuscript should be published in JoF.

Remarks

Line 37 oil[1]. Please leave a space (applies to the entire manuscript).

Line 80: How were Colletotrichum gloeosporioides and Alternaria alternata identified? If sequences were performed, it would be helpful to provide the GenBank accession number.

Line 89 in section 2.2 'Pathogen Isolation and Pathogenicity Assays' does not state that the pathogenicity test was performed only with the Colletotrichum siamense isolate; C. gloeosporioides and A. alternata were not tested.

Line 96 'Surface sterilized leaves were inoculated' should be supplemented to indicate whether these were leaves on living seedlings or leaves picked from trees and inoculated in the laboratory. It is likely that it was the latter.

Line 100 – after how long were symptom evaluations and re-isolation performed (information is found only in Results)?

Line 121-145: Check whether this text is consistent with the title ‘Phylogenetic reconstruction’.

Line 196: Mature leaves – provide more details, detached leaves?, origin, number.

Line 200: ‘Lesions were measured post-decoloration (75% ethanol),’ – the text requires clarification (more details are required).

Line 209: ‘A randomized complete block design with three replicates’ – the text requires further clarification, specifying how many trees were in each block.

Line 219: ‘Optimization of Field Application Strategies’ – were only fruits evaluated in this experiment (see line 230) – why isn't this stated at the beginning of the chapter? There was information about leaf analysis earlier.

Line 246-247 is the pathogenicity test performed on one leaf sufficient to assess the pathogenicity of the fungus?

Line 305: There's an error in the fungus name; It should be A. alternaria instead of A. alterica.

Line 305 There is an error in the fungus name; it should be C. siamense instead of C. siamensee.

Line 343 There is an error in the fungus name; it should be C. siamense instead of C. siamensee.

Line 343 There is an error in the fungus name; it should be A. alternaria instead of A. alterica.

Line 429 There is an error in the fungus name; it should be Botrytis cinerea instead of Botrytis cinereal

Line 562 Colletotrichum – it should be italic

Line 564 Juglans regia – it should be italic

Line 567 Juglansregia - it should be Juglans regia

Line 604 Verticillium dahliae – it should be italic

Line 606 Bacillus amyloliquefaciens – it should be italic

Round 2

Reviewer 1 Report

Dear authors, since all doubts and recommendations to improve the manuscript have been addressed, the document may be published.

My final recommendation would be to verify the entire document once you’ve applied the changes with the Word editor.